# Targeting Ultrasmall Gold Nanoparticles with cRGD Peptide Increases the Uptake and Efficacy of Cytotoxic Payload

**DOI:** 10.3390/nano12224013

**Published:** 2022-11-15

**Authors:** Richard D. Perrins, Lee-Anne McCarthy, Angela Robinson, Kelly L. Spry, Valentin Cognet, Avelino Ferreira, John Porter, Cristina Espinosa Garcίa, Miguel Ángel Rodriguez, Diana Lopez, Ibon Perera, Kelly Conlon, Africa Barrientos, Tom Coulter, Alessandro Pace, Sarah J. M. Hale, Enrico Ferrari, Csanad Z. Bachrati

**Affiliations:** 1Midatech Pharma Plc, 1 Caspian Point, Caspian Way, Cardiff CF10 4DQ, UK; 2School of Life Sciences, University of Lincoln, Joseph Banks Laboratories, Green Lane, Lincoln LN6 7DL, UK

**Keywords:** ultra-small gold nanoparticles, DM1, cRGD-peptide, αVβ3 integrin, targeted drug delivery

## Abstract

Cyclic arginyl-glycyl-aspartic acid peptide (cRGD) peptides show a high affinity towards αVβ3 integrin, a receptor overexpressed in many cancers. We aimed to combine the versatility of ultrasmall gold nanoparticles (usGNP) with the target selectivity of cRGD peptide for the directed delivery of a cytotoxic payload in a novel design. usGNPs were synthesized with a modified Brust-Schiffrin method and functionalized via amide coupling and ligand exchange and their uptake, intracellular trafficking, and toxicity were characterized. Our cRGD functionalized usGNPs demonstrated increased cellular uptake by αVβ3 integrin expressing cells, are internalized via clathrin-dependent endocytosis, accumulated in the lysosomes, and when loaded with mertansine led to increased cytotoxicity. Targeting via cRGD functionalization provides a mechanism to improve the efficacy, tolerability, and retention of therapeutic GNPs.

## 1. Introduction

Gold nanoparticles (GNPs) provide a useful scaffold to which a variety of different ligands may be attached to imbue desirable properties such as improved solubility, stability, stealth, delivering a therapeutic payload or targeting the GNP to a specific cell type or location through passive mechanisms (by enhanced permeability and retention effect) or active delivery (using a ligand specific to a tissue or a cell type) [1,2]. Targeting may lead to improved biodistribution, reduced drug degradation, prevention of off-target effects, and thereby increasing the therapeutic window [3]. The cyclic-peptide cRGD is a well-characterized integrin-binding ligand that has previously been used to redirect drugs [4], polydopamine nanoparticles [5], iron oxide nanoparticles [6], virus-derived particles [7], fluorescent polymer nanoparticles [8] to cells that express RGD motif interacting integrins: α8β1, αVβ3, αVβ5, αVβ6, αVβ8, and αIIbβ3 [9,10]. Several studies designed for in vitro and in vivo cancer cell imaging have confirmed the potential of such functionalized GNPs [11,12,13,14] and called for further characterization and development.

Composed of an inert gold core, GNPs are not expected to have cytotoxicity; consequently, in vivo toxicity data of GNPs is limited [15,16,17]. usGNPs are mainly cleared through the kidneys [18,19,20], while those particles that are not cleared accumulate in organs rich in elements of the mononuclear phagocyte system (MPS) such as the liver and spleen [20,21,22]. The core size was shown to affect systemic toxicity in vivo [17], however, surface charge and functionalization are thought to contribute more significantly to cytotoxicity [23,24,25,26].

Ultrasmall GNPs offer several advantages compared to GNPs with sizes over 10 nm. usGNPs have an improved surface/volume ratio, which allows the immobilization of larger amounts of payload at the same dose of gold. They have improved penetration and retention in tumors and fast clearance in normal tissues [27,28]. Similar to the long-recognized suitability of GNPs for delivering cytotoxic payload [29], such potential of usGNPs is also promising; delivery of drugs such as daunorubicine [30], gemcitabine [31], 6-mercaptopurine [32], dodecylcysteine [33], 5-fluorouracil [34], cisplatin [35,36], and doxorubicin [37] have been confirmed. Small GNPs also show greater X-ray attenuation than their larger counterparts, making them better suited as a base for CT contrast agents [38,39]. Furthermore, usGNPs proved to be a suitable nanocarrier for nuclear delivery, with a cut-off size of <10 nm to penetrate the nuclear pores [40]. Besides biocompatibility and tunable functionalization, the size of GNPs can be precisely controlled [41], thus providing an ideal platform to develop multi-functional nanocarriers with effective uptake into tissues, cells, and sub-cellular compartments.

In this study, we present evidence for specific, high-affinity physical interaction between cRGD-functionalized usGNPs and αVβ3 integrin in vitro, which mediates a selective uptake by cells that express αVβ3 integrin. We provide evidence that our cRGD-functionalized usGNPs are internalized via the clathrin-dependent endocytosis pathway and accumulate in the endosomes. We generated cytotoxic usGNPs by attaching mertansine (DM1), a tubulin polymerization inhibitor maytansinoid drug to the cRGD functionalized usGNPs. DM1 can be attached to usGNPs through its thiol group, is released via ligand exchange with intracellular reduced thiols such as glutathione [42], and has been commercialized in an antibody-drug conjugate therapy (Ado-trastuzumab emtansine, Kadcyla^TM^) [43]. Our data provide evidence that usGNPs functionalized with cRGD and DM1 are selectively cytotoxic to cells that express αVβ3 integrin and highlight the potential of the platform for the targeted delivery of therapeutic payloads, warranting further preclinical evaluation.

## 2. Materials and Methods

### 2.1. Base GNP Synthesis

Synthesis was performed with a modified Brust-Schiffrin method, using a Atlas Potassium reactor (Syrris Ltd., Royston, United Kingdom) with a 2 L jacketed torispherical vessel and a 500–50 mm blade propeller stirrer. The reaction was carried out at 18 °C with fast stirring (750 rpm). Time, pH, and temperature were continuously monitored. H_2_O (Ultrapure, MilliQ; MilliporeSigma, Burlington, VT, USA) was the solvent used for both synthesis and purification. The different reagents were added from the top of the vessel within minutes (~15 min) using a 150–80–8 mm funnel starting with HAuCl_4_•3H_2_O (1 equivalent, 400 mg, 1.016 mmol, 1645 mL; Sigma-Aldrich, St. Louis, MI, USA). Aqueous NaOH solution (~3 mL 2 M NaOH, Sigma-Aldrich) was used to modulate the pH to ~11.0. Freshly prepared NaBH_4_ in excess (0.2 M in 0.01 M NaOH, five equivalents, 192 mg, 5.078 mmol, 25 mL; AppliChem GmbH, Darmstadt, Germany) was quickly added to initiate the nucleation of the particles. Five seconds after nucleation, a 20 mL solution containing the disulfide ligands α-Thio-ω-(propionic acid) octa (ethylene glycol) (0.129 equivalent, 118 mg, 0.129 mmol, 10 mL; Acadechem Company Ltd., Hong Kong) and 2′-Thioethyl α-D-galactopyranoside (0.023 equivalent, 11 mg, 0.023 mmol, 10 mL; GalChimia, A Coruña, Spain) was added in an 85:15 molar ratio. Thirty seconds after nucleation, extra aqueous NaOH solution (~7 mL 2 M NaOH) was used to adjust the pH to ~12. After all reagents were added, the stirring speed was dropped to 550 rpm and the reaction was left to incubate for 30 min. The gold concentration was 0.6 mM and the final volume of the reaction corresponded to 85% of the reactor capacity. Purification was performed with a KR2i TFF system (Repligen, Waltham, MA, USA) using a hollow fiber filter with a 10 kDa molecular weight cut-off. The final product was filtered with a 0.22 μm membrane and resuspended in H_2_O at a final target gold concentration of 4 mg/mL.

### 2.2. Functionalization of GNPs

#### 2.2.1. Ligand Exchange of DM1 onto GNP

Base GNP (15 mg Au) was solvent exchanged into 38% (*v*/*v*) DMSO/H_2_O using 10 kDa Amicon 15-Ultra filters (MilliporeSigma) to achieve a final gold concentration of 1 mg/mL. To the 15 mL GNP solution, DM1 (30 equivalents per GNP, 843 µg, 1.142 µmol, 843 µL; WuXi AppTec, Shanghai, China) in DMSO (Sigma-Aldrich) was added. The reaction was stirred at 800 rpm for 3 h at room temperature. After completion of the reaction, the solution was diluted to 10% (*v*/*v*) DMSO in H_2_O. Purification was performed using 10 kDa Amicon 15-Ultra filters. The final product was filtered with a 0.22 µm membrane and resuspended in H_2_O at a final target gold concentration of 5 mg/mL [18].

#### 2.2.2. Peptide Functionalization of usGNPs

The coupling reagents Ethyl-3-(3-dimethylaminopropyl)carbodiimide (EDC, 3 equivalents per PEG(8)COOH, 8.5 mg, 44.162 µmol, 3.0 mL, Sigma-Aldrich) and N-Hydroxysuccinimide (NHS, six equivalents per PEG(8)COOH, 10.2 mg, 88.325 µmol, 3.0 mL, Sigma-Aldrich) in H_2_O were mixed and added to a GNP solution (40 mg Au, 2 mg/mL). The reaction was stirred at 600 rpm for 2 h at room temperature. A purification step was performed with 10 kDa Amicon 15-Ultra filters to remove the excess coupling reagents and the activated GNP was resuspended in 20 mL 1X phosphate buffered saline (PBS; Thermo Fisher Scientific, Waltham, MA, USA). cRGD-NH_2_ or cRAD-NH_2_ (WuXi AppTec) in 20 mL of 1X PBS were quickly added to the activated nanoparticles in two different amounts each (1.3–6.6 mg, 1.472–7.360 µmol) to obtain a high and a low loading (0.1 and 0.5 equivalents per PEG(8)COOH). The reaction was stirred at 600 rpm overnight at room temperature. Purifications were performed with 10 kDa Amicon 15-Ultra filters. The final products were filtered with a 0.22 µm membrane and resuspended in H_2_O at a final target gold concentration of 5 mg/mL [44].

#### 2.2.3. Fluorophore Functionalization of GNPs

The coupling reagents Ethyl-3-(3-dimethylaminopropyl)carbodiimide (EDC, three equivalents per PEG(8)COOH, 2.5 mg, 13.249 µmol, 1.5 mL, Sigma-Aldrich) and N-hydroxysulfosuccinimide (Sulfo-NHS, six equivalents per PEG(8)COOH, 5.8 mg, 26.497 µmol, 1.5 mL, (Thermo Fisher Scientific) in H_2_O were mixed and added to a GNP solution (12 mg Au, 2 mg/mL). The reaction was stirred at 600 rpm for 2 h at room temperature. A purification step was performed with 10 kDa Amicon 15-Ultra filters to remove the excess of coupling reagents and the activated GNP was resuspended in 20 mL 1X PBS. Sulfo-Cyanine5 amine (Fluoroprobes, Scottsdale, AZ, USA) in 80 µL of DMSO was quickly added to the activated particle (0.05 equivalents per PEG(8)COOH, 0.16 mg, 0.221 µmol). The reaction was stirred at 600 rpm overnight at room temperature. Purifications were performed with 10 kDa Amicon 15-Ultra filters. The final products were filtered with a 0.22 µm membrane and resuspended in H_2_O at a final target gold concentration of 5 mg/mL.

### 2.3. Elemental Analysis; Microwave Plasma—Atomic Emission Spectrometry (MP-AES)

The gold concentration of the synthesized usGNPs in H_2_O was measured using 4200 MP-AES with MP Expert software version 1.5.16821 (Agilent, Santa Clara, CA, USA). Samples were digested with *Aqua regia*, then diluted using 3% (*v*/*v*) HCl. 

### 2.4. Particle Sizing

#### 2.4.1. UV-Vis Spectra

Spectra (λ 350–700 nm) were obtained using Lambda 35 UV-Vis Spectrophotometer (PerkinElmer, Waltham, MA, USA) with a quartz cuvette (20 μg/mL Au, diluted in water). 

#### 2.4.2. Transmission Electron Microscopy (TEM)

Samples were prepared under ambient conditions by desiccating a 0.35 μL droplet of a 150 μg/mL Au GNP aqueous solution on a hydrophilized carbon film surface. Ultrathin film supports type #01824 (Ted Pella Inc., Redding, CA, USA) were used. The hydrophilization was performed for 2 min with 25 mA strong glow discharge treatment in a K100X plasma chamber (Quorum Technologies Ltd., Laughton, United Kingdom). Images were acquired in a transmission electron microscope of type JEM-2100F [Model EM-20014, UHR, 200 kV] (JEOL, Tokyo, Japan) equipped with a digital camera of type F-216 (TVIPS, Gauting, Germany). Wider field-of-view images at X150k magnification were assembled with the spotscan utility of the TVIPS EMMENU4 software from a beam shift based 4 × 4 images matrix. Analyses were performed at CIC BiomaGUNE (San Sebastian, Spain). Data processing was performed using ImageJ (National Institutes of Health, Bethesda, MD, USA).

#### 2.4.3. Dynamic Light Scattering (DLS)

Size (hydrodynamic diameter) measurements were performed with Zetasizer Nano-ZS (Malvern Panalytical Ltd., Malvern, United Kingdom). Size was expressed by volume (%) as an average of three measurements taken at 25 °C at a 173-degree scattering angle in a plastic cuvette (200 μg/mL Au, diluted in 10X PBS).

#### 2.4.4. Differential Centrifugation Sedimentation (DCS)

Sizing analysis was performed using a CPS DC24000UHR disc centrifuge (CPS Instruments Inc., Prairieville, LA, USA) as described previously [18]. Eleven milliliters of 8–24% sucrose gradient was made up in water. Solutions with decreasing sucrose concentration were injected sequentially to create the gradient. Gradient evaporation was reduced by the injection of 500 μL dodecane. Prior to data acquisition, the gradient was allowed to reach thermal equilibrium and stabilize for about 30 min. A total of 0.237 μm polyvinylchloride (PVC) calibration standards in 50 μL injection volume were analyzed prior to each GNP sample (100 μL, 100 μg/mL Au) to ensure that the instrument was operating with a high degree of accuracy and optimally. Analyses were carried out at 24,000 rpm with the light detector adjusted to a position optimal for the analysis of usGNPs. Particle size was calculated based on an assumed GNP density of 5.0 g/cm^3^. Size was expressed by surface mode. 

### 2.5. Ligand Ratio; ^1^H NMR

The GNP amount equivalent to 10 mg of Au was incubated with 600 µL of 0.3 M KCN in 0.1 M KOH (solvent D_2_O) after the removal of H_2_O by freeze-drying. Particles were incubated at 80 °C for 6 h with strong agitation to prevent pelleting (950 rpm). A transparent solution with no pellet indicated complete etching, which was visually checked. Experiments were performed at 298 K on a AVANCE III 500 spectrometer (Bruker, Billerica, MA, USA) at CIC BiomaGUNE (San Sebastian, Spain) (500 MHz, D_2_O). Data processing was performed using MestReNova 10.0.2 (Mestrelab Research S.L, A Coruña, Spain).

### 2.6. UHPLC-CAD-MS

The GNP amount equivalent to 350 μg of Au was incubated with 15 μL of 0.3 M KCN and 0.01 M KOH and H_2_O up to 190 μL. Mixing was carried out by vortexing. Particles were incubated at 80 °C for 10 min with strong agitation to prevent pelleting (950 rpm). It was considered to have complete etching when, visually, a transparent solution with no pellet was attained. To the etched solution, 10 μL of 0.05 M TCEP that was made from a commercial, neutral 0.5 M solution (aqueous solution; pH 7.0 adjusted with ammonium hydroxide, Sigma-Aldrich) was added. 

UltiMate 3000 Rapid Separation Liquid Chromatography system (Thermo Fisher Scientific) comprising a dual gradient standard pump, Corona Veo RS CAD detector (Chromeleon 7.0 software) in line with LCQ Fleet Ion trap Mass Spectrometer detector (Xcalibur 2.2 SP1 software) was used for all experiments. Separation steps were performed on an Acquity UPLC BEH C18 column, 130 Å, (100 × 2.1 mm i.d., 1.7 µm particle size) and an Acquity UPLC BEH C18 VanGuard precolumn, 130 Å, (5 × 2.1 mm i.d., 1.7 µm particle size) (Waters, Milford, CT, USA). Solvents used as mobile phase were A: 0.1% formic acid in H_2_O; B: 0.1% formic acid in acetonitrile. Elution conditions: 0–0.5 min, 5% B isocratic; 0.5–6 min, linear gradient 5–98% B; 6–7 min, 98% B isocratic; washing and reconditioning of the column. The flow rate was 0.350 mL/min and the injection volume was 5 µL. The system operated at 35 °C. The Corona Veo RS Evaporation temperature was set at 35 °C; Power function: 1.0; Data collection Rate: 2 Hz; Signal Filter: 3.6 s. ESI-MS analysis was performed in the positive ion mode. Nitrogen was used as a desolvation gas. The ESI parameters of the source were: capillary temperature of 150 °C, the source heater temperature was held at 45 °C, and a potential of 3.8 kV was used on the capillary for positive ion mode. MS spectra, within the *m*/*z* range of 150–2000 amu, were obtained at 35 V cone voltage.

### 2.7. HPLC-MS

The GNP amount equivalent to 12 µg of Au was incubated with 60 µL of 0.5 M TCEP with 50% (*v*/*v*) DMSO/H_2_O up to 120 µL. Mixing was carried out by vortexing. Particles were incubated at 80 °C for 1 h with strong agitation to prevent pelleting (950 rpm). 1260 Infinity system (OpenLab CDS software) in line with 6120 Single Quadrupole mass spectrometer was used for all experiments (Agilent). Separations were performed on an Ascentis Express Peptide C-18 octadecyl phase column (4.6 × 100 mm, 2.7 μm) and an Ascentis Express C18 octadecyl phase guard column (4.6 × 5 mm, 2.7 µm (Sigma-Aldrich). Solvents used as mobile phase were A: 0.1% trifluoroacetic acid in H_2_O; B: 0.1% trifluoroacetic acid in acetonitrile for LC-DAD; A: 0.1% acetic acid in H_2_O; B: 0.1% acetic acid in acetonitrile for LC-MS. Elution conditions: 0–2 min, 20% B isocratic; 2–8 min, linear gradient 20–100% B; 8–9 min, 100% B isocratic; 9–10 min linear gradient 100–20% B; 10–12 min, 20% B isocratic; washing and reconditioning of the column. 

The flow rate was 1 mL/min and the injection volume was 10 µL. The system operated at 35 °C. ESI-MS analysis was performed in the positive ion mode. 

### 2.8. Octet Binding Studies

αVβ3 integrin was biotinylated by reacting 50 µL 1 mg/mL αVβ3 integrin (R&D systems, Minneapolis, MN, USA) with 5.2 µM EZ-Link NHS-PEG_4_-Biotin (Thermo Fisher Scientific) in PBS at room temperature for 1 h. The reaction was stopped by purification on a Zeba spin 7 kDa MWCO desalting column (Thermo Fisher Scientific) following the manufacturer’s protocol [45].

Binding interactions were measured using an Octet Red 96 (Sartorius, Göttingen, Germany) with High Precision Streptavidin (SAX) biosensors as per the manufacturer’s guidelines. In brief, biosensors were hydrated in binding buffer (50 mM HEPES, pH 7.4, 150 mM NaCl, 5 mM KCl, 0.1% (*w*/*v*) Bovine Serum Albumin (BSA; Sigma-Aldrich), 0.05% (*v*/*v*) Tween 20, 50 µM CaCl_2_) for >10 min, then washed twice in binding buffer (1 min then 3 min). Ten nanometers of biotinylated-αVβ3 integrin in binding buffer was loaded onto the biosensor for 15 min, quenched with 300 µM biocytin (Thermo Fisher Scientific) in binding buffer for 2 min, washed with binding buffer for 1 min, and the baseline in binding buffer measured for 3 min. αVβ3-loaded biosensors were allowed to associate with usGNPs in binding buffer for 40–60 min and dissociate in the buffer used for baseline measurements or in 1 µM cRGD-peptide in binding buffer for 40–60 min. To control for biosensor drift due to the dissociation of the αVβ3 monomers, measurements were referenced by subtracting the response from an αVβ3-loaded sensor with only a binding buffer in the association step. All measurements were taken at 30 °C. 

The dissociation constant for the interaction was approximated by plotting the sensorgram response at equilibrium (averaged response at 3570–3600 s) against the nanoparticle concentration (assuming negligible depletion of the concentration of free usGNPs at equilibrium, [usGNP]_free_ ≈ [usGNP]_total_). Results were normalized to give a percentage bound, to overcome any possible variation in R_max_ (maximum response) due to the use of different batches of biotinylated-integrin and biosensors. Binding data were analyzed using Data Analysis HT 10.0.1.7 (Sartorius). R was used to fit the equilibrium binding data to the Hill equation below using non-linear least square regression to estimate the dissociation constant (*K_d_*) and the Hill coefficient (*h*).
(1)$$Fraction\;Bound=\frac{{\left[usGNP\right]}^{h}}{K_{d}+{\left[usGNP\right]}^{h}}$$

### 2.9. Cell Lines and Routine Maintenance

U-87 MG [46], Hep3B, and HEK-293 cells were obtained from ATCC (Manassas, VA, USA). The U-251 MG cell line was a kind gift from Dr Cinzia Allegrucci at the University of Nottingham. The U-87 MG, Hep3B, and U-251 MG cells were routinely cultured and treated in EMEM (Sigma-Aldrich) supplemented with L-glutamine (Sigma-Aldrich), 10% (*v*/*v*) FBS (ATCC) non-essential amino acids (Sigma-Aldrich) and sodium pyruvate (Sigma-Aldrich). HEK-293 cells and their derivatives (see below) were maintained in low glucose DMEM (Thermo Fisher Scientific) supplemented with L-Glutamine, sodium pyruvate, and 10% (*v*/*v*) FBS (Thermo Fisher Scientific). Cells were kept at 37 °C, in a humidified atmosphere containing 5% CO_2_, unless stated otherwise.

### 2.10. Generation of αVβ3 Integrin Expressing HEK-293 Cell Lines

HEK-293 cells were transfected with a plasmid harboring a Myc-DDK tagged CD61 (β3 integrin) cDNA (RC221606; OriGene, Rockville, MD, USA). The transfectants were selected with 500 µg/mL G418 (Geneticin; Thermo Fisher Scientific). Surviving colonies were analyzed for expression with anti-FLAG immunofluorescence; the highest expressors with normal morphology were chosen for subsequent analysis. Generation and initial screening of clones were carried out by SAL Scientific Ltd. (Fordingbridge, UK).

### 2.11. Cellular Uptake of usGNPs

Cells were plated, at 2 × 10^6^ cells/well into collagen (10 µg/mL) coated 6-well plates (Eppendorf, Hamburg, Germany) and cultured overnight. The medium was exchanged for 400 µL/well fresh low serum (2% [*v*/*v*] FBS) EMEM containing usGNPs at 1 × 10^8^ particles per cell (ppc) and incubated for 60 min as normal or at 4 °C. Competition experiments were preincubated with 500 µM free cRGD-peptide in 300 µL/well low serum EMEM for 60 min, then the usGNP solution was added at 1 × 10^8^ ppc in a 100 µL volume (final volume 400 µL/well) and incubated for 60 min. 

Post-treatment with usGNPs, the cells were scraped from the plates, transferred to 2 mL tubes, and centrifuged at 200× *g* for 5 min. The supernatant was discarded, and the pellet was washed twice with 1 mL PBS, then once with 1 mL 0.2 M acetic acid (pH 2.8) in 0.5 M NaCl at 4 °C, to remove surface-bound material. Samples were centrifuged at 200× *g* for 5 min and washed a further time with 1 mL PBS and stored at −20 °C prior to analysis by ICP-MS.

Cell pellets were lysed in 1.6 mL 3% (*w*/*v*) tetramethylammonium hydroxide (TMAH; Sigma-Aldrich) solution containing 0.2% (*v*/*v*) Triton ×100 (Sigma-Aldrich) under agitation for approximately 30 min. A total of 1.55 mL of lysate was transferred into a new tube and topped up with 1 mL 3% (*w*/*v*) TMAH. Internal standard solution (2.45 mL of 4 ppb iridium (Sigma-Aldrich), 3% (*v*/*v*) HCl in H_2_O) was added to the samples and the resulting solution was measured on a NexION 300× ICP-MS instrument (PerkinElmer) and quantitated using a calibration curve (Figure 4). Alternatively, usGNP uptake was measured using inductively coupled plasma—atomic emission spectroscopy (ICP-AES, Figure 5) using the iCAP 7400 ICP-AES Analyzer (Thermo Fisher Scientific) [47].

### 2.12. Cell Viability

Cells were plated at 2 × 10^4^ cells/well for Hep3B cells and 1 × 10^4^ cells/well for U-87 MG cells into 96-well plates (Eppendorf) and cultured overnight. The medium was exchanged for 200 µL/well phenol red-free EMEM (Thermo Fisher Scientific) supplemented with 10% (*v*/*v*) FBS, non-essential amino acids, and sodium pyruvate containing an 8-point 3-fold dilution series of compound or usGNP in triplicate. Cells were then incubated for 3-days as normal. Absorbance reading at 475 nm without pathlength correction was taken from all wells prior to the addition of XTT and PMS as follows. To each well of the plate 50 µL 1 mg/mL 2,3-Bis-(2-Methoxy-4-Nitro-5-Sulfophenyl)-2H-Tetrazolium-5-Carboxanilide (XTT, Thermo Fisher Scientific) and 10 µM phenazine methosulfate (PMS, Thermo Fisher Scientific) in phenol red-free EMEM was added. Cells were then incubated for 3 h as normal. Absorbance at 475 nm without pathlength correction was measured with SPECTROstar nano microplate reader (BMG Labtech, Ortenberg, Germany) and the corresponding A_474nm_ reading obtained before XTT/PMS incubation was subtracted. The subtracted data were normalized and an IC_50_ was obtained by fitting a four-parameter [Inhibitor] vs. response curve using Prism 9 (GraphPad Software, San Diego, CA, USA) [48].

### 2.13. Indirect Immunofluorescent Microscopy

Cells were seeded at 1 × 10^5^ in wells of a 24-well plate with glass coverslips, incubated overnight then treated as indicated. The medium was removed and the coverslips were rinsed with PBS twice. Cells were fixed on ice for 10 min with 4% paraformaldehyde in 250 mM HEPES pH 7.4 (both from Sigma-Aldrich). For detecting intracellular antigens cells were permeabilized with 0.1% Triton X-100 (Thermo Fisher Scientific) in PBS for 20 min on ice. Coverslips were then incubated for 1 h at 37 °C in a blocking buffer (10% FBS, 0.1% Triton X-100 in PBS). Coverslips were then placed in a humidity chamber and incubated overnight at 4 °C with 15 µL of primary antibody solution in blocking buffer diluted as follows: mouse anti-human integrin β3 primary antibody, 1:200 (11–0519-42; Thermo Fisher Scientific); rabbit anti-EEA1, 1:200 (3288; Cell Signaling Technology, Danvers, MA, USA); rabbit anti-RAB7, 1:100 (9367; Cell Signaling Technology); rabbit anti-RAB11, 1:100 (5589; Cell Signaling Technology); rabbit anti-RCAS1, 1:100 (12290; Cell Signaling Technology). Coverslips were washed 4× for 10 min with PBS with gentle rocking, then were subsequently incubated with 100 µL of Alexa Fluor 488 conjugated goat anti-mouse secondary antibody (1:800 dilution, A11001; Thermo Fisher Scientific) or Alexa Fluor 488 conjugated goat anti-rabbit secondary antibody (1:800 dilution, A11008; Thermo Fisher Scientific) at 37 °C for 1 hr. The coverslips were washed 4× with PBS as above, then with ultrapure water, dried, and mounted onto microscope slides using Vectashield^®^ mounting medium containing DAPI (Vector Laboratories Inc., Newark, NJ, USA). Imaging was performed on a TCS SP8 laser scanning microscope with an HC PL APO 63×/1.40 Oil CS2 objective (Leica Microsystems, Wetzlar, Germany). Image acquisition and processing parameters were kept identical between samples from the same experiment to ensure comparability.

### 2.14. Flow Cytometry

Cells were seeded at 1 × 10^4^ per well in a 6-well plate and incubated for 24 h. Cells were detached using 0.5% trypsin (Thermo Fisher Scientific), suspended in 2 mL of PBS then collected in a conical centrifuge tube. Samples were centrifuged at 220× *g* for 3 min, then suspended in 1 mL PBS + 0.3% BSA (Sigma-Aldrich). All samples were filtered using a 100 µm Corning cell strainer (Thermo Fisher Scientific). Cells were counted using TC20^TM^ Automated Cell Counter (Bio-Rad Laboratories, Hercules, CA, USA). Cell density was set to 1 × 10^6^, and 25 µL of cell suspension was then incubated with 25 µL FITC pre-conjugated primary antibodies (mouse anti-human integrin β3; 11–0519-42, 1:200; Thermo Fisher Scientific) or isotype control antibodies (mouse IgG kappa isotype control; 11–4714-42, 1:200; Thermo Fisher Scientific) for 30 min at room temperature in U bottom 96 well plates. Samples were centrifuged at 260× *g* for 2 min. Cells were then washed three times in 100 µL PBS + 0.3% BSA, then resuspended in 200 µL PBS + 0.3% BSA. The samples were transferred into a FACS tube with 200 µL FACS sheath fluid. Cells were analyzed by flow cytometry using a FACSVerse flow cytometer (Becton, Dickinson and Company, Franklin Lakes, NJ, USA). A minimum of 10,000 cells were analyzed per sample. Data were collected and analyzed using FACSuite and FlowJo software packages (Becton, Dickinson and Company). 

### 2.15. Live Cell Imaging

Cells were seeded in 35 mm µ-dish (ibidi) or Nunc Lab-Tek II 8-well coverglass (Thermo Fisher Scientific) vessels at 2.85 × 10^4^ cells per cm^2^, incubated overnight then treated as indicated in the figure legends with chlorpromazine-hydrochloride (Sigma-Aldrich), genistein (Sigma-Aldrich), or nocodazole (Sigma-Aldrich). Live cell imaging was performed on a TCS SP8 laser scanning confocal microscope with an HC PL APO 63×/1.40 Oil CS2 objective. Cells were stained as indicated in the figure legends with BODIPY™ FL C5-Lactosylceramide BSA complex (Thermo Fisher Scientific) or LysoTracker^®^ Red (Thermo Fisher Scientific). Multiple dyes were imaged with sequential acquisition settings to ensure a clear signal from single dyes without bleed-through into other acquisition channels.

## 3. Results and Discussion

### 3.1. Synthesis and Characterization of Functionalized Gold Nanoparticles

In order to characterize the uptake of ultrasmall gold nanoparticles functionalized with cRGD and to prove that they can deliver cytotoxic cargo to cells, plasmonic GNPs with 4 nm cores were synthesized using a modified aqueous Brust-Schiffrin method [49,50,51]. Two types of ligands were used in a 50:50 ratio to make base usGNPs: a monosaccharide with a short ethyl side chain 2′-Thioethyl α-D-galactopyranoside (α-Galactose-C_2_) and an oligoethylene glycol α-Thio-ω-(propionic acid) octa(ethylene glycol) (PEG(8)COOH). Two-nanometer core GNPs functionalized with the same mixed corona and the SIKVAV peptide have been previously demonstrated to have the ability to target cancer cells through α6β1 integrins [52]. A similar structure with a 2 nm core functionalized with cRGD and the same negatively charged oligoethylene glycol ligand also showed promising results in vitro [53]. Carbohydrates are known for their ability to improve stability (avoid aggregation), solubility, biocompatibility, biodegradability and confer stealth (protein-repellent) properties [54]. Polyethylene glycols (PEG), among them oligoethylene, are flexible molecules, relatively inert (non-immunogenic), and soluble in water, as well as most polar organic solvents. They possess a strong ability to stabilize particles by preventing nanoparticle aggregation [55,56]. They also improve the characteristics of the nanoparticles in vitro and in vivo, such as half-life or oral bioavailability [57]. The terminal functional group of PEG can additionally be used to bind to molecules presenting a complementary moiety to achieve active targeting [2]. According to the model of Vergara et al. [58], the 4 nm GNPs are composed of an average of 2000 gold atoms and 290 ligands and henceforth depicted as (Ligands)_290_@Au_2000_. The base particle obtained through the modified Brust-Schiffrin synthesis: (α-Galactose-C_2_)_145_(PEG(8)COOH)_145_@Au_2000_ was then functionalized using two strategies: post-functionalization and ligand exchange. Post-functionalization was performed by amidation of the carboxyl-terminal moiety of the oligoethylene glycol ligand with amine derivatives of cRGD, cRAD (similar structure but with lower affinity to αVβ3 integrin) [59,60], and the Sulfo-Cyanine5 amine fluorophore [61,62]. Ligand exchange strategy was used to load DM1 [18,63] (Figure 1).

The post-functionalization synthetic procedure permitted the control of the number of cRGD moieties loaded per usGNP using simple stoichiometric variations of a cRGD amine derivative (cRGD-NH_2_). Consequently, two different versions were synthetized: _High_cRGD-usGNP and _Low_cRGD-usGNP with approximately 60 and 20 cRGD units per usGNP, respectively. The same method was applied to obtain _High_cRAD-usGNP and _Low_cRAD-usGNP. The base usGNPs and _Low_cRGD-usGNPs were functionalized with approximately 20 DM1 per GNP. 1–2 Sulfo-Cyanine5 amine was added to the usGNPs for fluorescent studies (Cy5-usGNP) (Table 1).

Different analytical techniques were employed to characterize the core and overall size of the usGNPs. UV-Vis spectroscopy displayed a surface plasmon band (SPR) with a local maximum of around 520 nm (Figure 2a). Transmission Electron Microscopy (TEM) showed a mean core size of 4 nm (Figure 2b). Dynamic Light Scattering (DLS, Figure 2c, Table 1) and Differential Centrif ugation Sedimentation (DCS, Figure 2d) confirmed the monodispersity and that despite functionalizations, the particles remain in the ultrasmall (<10 nm) range (Table 1).

Proton Nuclear Magnetic Resonance (^1^H NMR, Appendix A) and Liquid Chromatography (LC) with different detectors (UV, Charged Aerosol Detector (CAD) or Mass Speedometer (MS)) were used to characterize the ligand corona. ^1^H NMR and LC analyses were performed after core etching and ligand release using a solution of potassium cyanide in potassium hydroxide (KCN/KOH) [64,65]. For LC applications, tris(2-carboxyethyl)phosphine (TCEP), a disulfide bond reducer, was added to prevent spontaneous thiol oxidation of the ligands after release from the core. A single method of LC-CAD-MS was used to determine the relative amounts of the base GNP ligands, cRGD, cRAD, and the fluorophore (Appendix A) by dividing the area of the peaks obtained with the CAD detector by the molecular weight of each ligand [66,67]. Absolute DM1 quantification was performed separately by LC-MS (λ 276 nm, Appendix A). The relative amounts (or absolute for DM1) of the ligands obtained by ^1^H NMR and LC analyses were then fitted in the previously mentioned model of Vergara et al. [58] with a total amount of 290 ligands per GNP (Table 1). The 1:1 ratio between α-Galactose-C_2_ and PEG(8)COOH in the corona of the Base usGNP was measured by both ^1^H NMR and LC-CAD. Equally, both techniques validated the high (60 per GNP) and low (20 per GNP) loading of the peptides. Sulfo-Cyanine5 amine loading was only measured by LC-CAD (1–2 Fluorophores per GNP). DM1 quantification in mg/mL was converted into equivalents per GNP (20 DM1 per GNP). The results presented above suggest a significant improvement on our previous GNP design based on a 2 nm core that can carry eight SIKVAV ligands per 200 gold atoms [52]. The slightly increased size still puts the 4 nm design in the ultrasmall range keeping its advantages outlined above, but it can support at least a 6x increase in cargo-bearing capacity.

### 3.2. Functionalized usGNPs Bind αVβ3 Integrin

αVβ3 integrin binding and release of cRGD functionalized usGNPs were measured with biolayer interferometry [45]. First, _High_cRGD-usGNP and control base usGNP were used to assess the specificity of binding to the immobilized αVβ3 integrin. _High_cRGD-usGNP showed robust binding with a K_d_ of 29.2 ± 3.2 pM (*p* < 0.01) and a Hill coefficient (h) of 1.11 ± 0.25 (*p* < 0.05). In the absence of cRGD, the binding of 1 nM control base usGNP to the immobilized αVβ3 integrin was unmeasurable, suggesting the absence of non-specific binding to αVβ3 integrin of any of the components of the base particle (Figure 3a). The very low K_d_ measured for _High_cRGD-usGNP suggests that avidity due to a large number of cRGD moieties may play a role in the binding, with little or no cooperativity (h is approximately 1). As expected, the _Low_cRGD-usGNP had a weaker affinity dropping approximately 100-fold (Figure 3b) with a K_d_ of 3.2 ± 0.4 nM (*p* < 0.001) and h of 0.44 ± 0.03 (*p* < 0.001). The negative cooperative binding highlighted by the Hill coefficient <1 is likely due to electrostatic repulsion between nanoparticles densely bound to the sensor, which becomes apparent at high usGNP concentrations. Importantly, the data on _Low_cRGD-usGNPs suggest that high-affinity binding (3.2 nM) is still possible with the lower cRGD loading, therefore leaving room for the conjugation of a drug payload.

The binding of the cRGD-usGNP to immobilized αVβ3 integrin was Ca^2+^ dependent as evidenced by the absence of binding in the presence of 10 mM EDTA (Figure 3c), which is a long-known property of the interaction between cRGD-peptide and αVβ3 integrin [68]. Figure 3c also shows that the dissociation of cRGD-functionalized usGNPs from the immobilized αVβ3 integrin was undetectable over 1 h. The addition of 1 µM cRGD-peptide to the dissociation buffer resulted in a reduction in response during the dissociation phase (Figure 3d), suggesting that the addition of the peptide is causing competitor-induced dissociation of the cRGD-usGNP from the immobilized integrin [69]. The slow dissociation rate of the cRGD-usGNP could be either due to an avidity effect from the multiple cRGD-ligands decorating the usGNP, and/or the high density of the gold-core reducing the rate of diffusion of the usGNP making rapid rebinding of the usGNP to the integrin more likely [69].

### 3.3. Functionalized usGNPs Are Taken up by Cells by an αVβ3 Integrin Mediated Mechanism

To test the uptake of cRGD functionalized usGNPs, pairs of cell lines with high and no or very low αVβ3 integrin expression but otherwise similar characteristics were chosen. The U-87 MG glioblastoma cell line [46] has been reported to highly express αVβ3 integrin [70,71,72,73,74], while the U-251 MG [71,73] is a limited αVβ3 expressor. While both these cell lines are of glioblastoma origin, they are genetically different, for which ectopic expression of the αVβ3 integrin was established in the HEK-293 human embryonic kidney cell line that does not normally express this receptor. Expression was confirmed by flow cytometry and immunofluorescent microscopy (Appendix A). The uptake of usGNPs was characterized and quantitated with elemental analysis of the gold content by inductively coupled plasma mass spectrometry (ICP-MS) and confocal microscopy using a set of usGNPs functionalized with Sulfo-Cyanine5 amine. 

Using the U-87 MG cell line, which expresses a high level of αVβ3 integrin, the uptake of usGNPs with different characteristics was quantified with ICP-MS and visualized with live cell confocal microscopy. While the uptake of the base usGNPs was minimal, a substantial amount of _Low_cRGD-usGNPs was taken up by the cells, which increased by ~3.4-fold from ~4.9 × 10^6^ to ~16.7 × 10^6^ usGNPs per cell by an increased density of cRGD peptides on the surface of the _High_cRGD-usGNPs (Figure 4a and Figure 5a–c). usGNPs, functionalized with cRAD peptide that has a low affinity to the αVβ3 integrin [75,76], showed a comparatively low uptake (~1.6 × 10^6^ and ~2.7 × 10^6^ for _Low_cRAD and _High_cRAD -usGNPs, respectively; Figure 4a). Pre-incubating the cells with 500 µM cRGD-peptide resulted in a reduction in usGNP uptake to a level (~2 × 10^5^ usGNPs per cell) similar to that found for the base usGNP (~10^5^ usGNPs per cell) confirming the target specificity (Figure 4b). Reducing the temperature to 4 °C resulted in a 7-fold loss in uptake from ~2.7 × 10^6^ to ~3.8 × 10^5^ usGNPs per cell (Figure 4c), confirming that uptake is via an active process.

The dependency of uptake on αVβ3 integrin expression by host cells was established using the verified cell line pairs and experiments using confocal microscopy and gold content by inductively coupled plasma atomic emission spectroscopy (ICP-AES). As expected, the uptake of cRGD-loaded usGNPs by the U-87 MG cell line was visible after 1 h, while no uptake was seen by the U-251 MG cell line (Figure 5c,d). Surprisingly, no uptake was seen by HEK-293 clone 5, a sufficiently high αVβ3 integrin expression which was confirmed earlier (Appendix A). ICP-AES confirmed uptake by U-87 MG cells. Furthermore, with this technique limited uptake by U-251 MG cells was also visible, in line with their limited albeit not zero αVβ3 integrin expression (Figure 5e). However, uptake by the parental HEK-293 or its αVβ3 integrin expressing derivative clone 5, could not be shown. Treatment of clone 5 cells with _High_cRGD-usGNPs induced their detachment from the substrate making detection and measurement of uptake unreliable (data not shown). HEK-293 cells express the αV integrin subunit, which forms the RGD binding integrin heterodimers pairing with a β subunit [10]. Of these, HEK-293 cells express β1, but not β3, β5 or β6 [77]. RGD binding integrins tend to have distinct subcellular localization patterns [78], which could be imbalanced by the ectopic expression of β3 together with the assembly and function of clathrin-coated structures. Furthermore, ectopic expression of the β3 subunit has also been reported to transform apoptotic signaling pathways characteristic to endothelial cells into epithelial HEK-293 cells [79]. These imbalances could explain the detachment of our _High_cRGD-usGNP treated clone 5 cells, nevertheless, we observed no unusual cell death of the untreated CD61-HEK-293 cells.

These data strongly indicate that the uptake of usGNPs is cRGD-dependent, possibly through an endocytic mechanism mediated by binding to the αVβ3 integrin.

### 3.4. usGNP Uptake Is Mediated through Clathrin-Dependent Mechanisms

Integrin-mediated endocytosis occurs via several distinct pathways [10]. In order to gain insight into which of these is utilized for the uptake of functionalized usGNPs, specific inhibitors of clathrin-dependent and -independent mechanisms were employed (Appendix A). We observed significant cytotoxicity using chlorpromazine-hydrochloride (CPH) at concentrations other laboratories employed. At the lowest concentration (2.5 µg/mL) distribution of endocytosed BODIPY™ FL C5-Lactosylceramide BSA complex (LaCer), which is endocytosed via the clathrin-independent route [80], was normal without signs of cytotoxicity; however, at higher concentrations membrane blebbing was observed, and the cytoplasm was filled with the signal from LaCer indicating compromised membrane integrity (Appendix A). Consequently, for subsequent experiments, 2.5 µg/mL CPH was used to inhibit clathrin-dependent endocytosis. Cytotoxicity of genistein, an inhibitor of clathrin-independent endocytosis was also tested, but no such effect was found in the applied concentration range (data not shown). Challenges of selective pharmacological inhibition of uptake mechanisms due to toxicity have also been highlighted by other laboratories [81].

To assess which endocytic pathway was responsible for the internalization of usGNPs, U-87 MG cells were pre-treated with CPH, genistein, and nocodazole, a microtubule poison, and inhibitor of intracellular trafficking [82], or left untreated, then incubated with _High_cRGD-usGNPs and LaCer together for 2 h. Uptake of _High_cRGD-usGNPs was inhibited by CPH but not by genistein, while uptake of the LaCer control was inhibited by genistein and not by CPH (Figure 6a–d). Nocodazole treatment moderately interfered with the internalization and intracellular trafficking of both _High_cRGD-usGNPs and LaCer (Figure 6e), similar to the reported, cell type-specific, effect found by other laboratories [82]. These results indicate that the _High_cRGD-usGNPs are internalized via the clathrin-dependent endocytic pathway. cRGDfK functionalized 50 nm GNPs have been reported to similarly internalize via clathrin-dependent routes, although sensitivity of uptake to the macropinocytosis inhibitor 5-(N-ethyl-N-isopropyl) amiloride (EIPA) was also found [70]. An uptake of non-targeted nanoparticles has been reported to be cell-type specific [81] and was also found to be dependent on surface properties [83], which highlights the importance of functionalization to target nanoparticles to specific receptors, such as cRGD targeting to αVβ3 integrin in our design.

### 3.5. Internalized usGNPs Accumulate in the Lysosomes

The moderate sensitivity of internalization to nocodazole prompted further analysis of intracellular trafficking of our functionalized usGNPs. Uptake and localization of Cy5 labeled _High_cRGD-usGNP were followed in live cell imaging in U-87 MG cells labeled with LysoTracker^®^ Red 15 min before microscopy. The presence of usGNPs in lysosomes, indicated by co-localization of the LysoTracker^®^ Red and Cy5 signals, became obvious after 30 min of incubation and reached a maximum after 120 min (Figure 7). Similar uptake kinetics of 4nm fluorescein-PEG-tagged usGNPs was observed in Raw264.7 macrophage cells with limited usGNP accumulation after 15 min that gradually increased after 1 h [84].

It was intriguing, however, that the co-localization of LysoTracker^®^ Red and _High_cRGD-usGNP was only partial at any time point, which suggested a dynamic process in which the lysosomes constitute a stage only. We attempted to follow the usGNPs outside the lysosomes by labeling endosomal compartments with antibodies against their respective specific markers [10]. Anti-EEA1 antibody was used for labeling early endosomes, anti-RAB7 for late endosomes, and anti-RAB11 for recycling endosomes. In addition, an anti-RCAS1 antibody was used to label the Golgi compartment. Unfortunately, we found that the usGNPs leached out of the cells during permeabilization that was included to enable diffusion of the antibody into fixed cells (data not shown). Wu and co-workers successfully used co-staining of their 15 and 20 nm GNPs with antibodies on fixed cells with a similar methodology [85]. Our unsuccessful attempt to co-stain the usGNPs with antibodies likely reflects the 4 nm size of the usGNP core, which is smaller than the size of the antibodies [86].

### 3.6. Dual-Functionalized usGNPs Show Selective Cytotoxicity on αVβ3 Integrin Expressing Cells

Lastly, we tested the ability of our functionalized usGNPs to deliver cytotoxic payload preferentially to αVβ3 expressing cells. Cytotoxicity of DM1 functionalized base usGNPs and _Low_cRGD-usGNPs was compared to DM1 alone on U-87 MG and Hep3B cells that do not express αVβ3 integrin [87]. usGNPs without DM1 were not cytotoxic over the same concentration range used for the functionalized usGNPs in either cell line (data not shown). For the Hep3B cell line, the cytotoxicity of DM1 was significantly reduced by the attachment to the usGNPs, presumably due to the rate of drug release from the usGNP. This is shown as an increase in the IC_50_ value from 3.2 nM (95% CI: 2.4–4.7 nM) for free DM1, to 30.1 nM (95% CI: 25–36.3 nM) equivalent concentration for DM1-loaded usGNP (Figure 8a). Functionalization of the DM1-usGNP with the cRGD peptide did not significantly alter the cytotoxicity of the particle in Hep3B cells (IC_50_: 30.4 nM; 95% CI: 26.8–34.5 nM). However, for the αVβ3 integrin expressing U-87 MG cell line (Figure 8b), cRGD functionalization of DM1-_Low_cRGD-usGNPs improved targeted cytotoxicity of the nanoparticle compared to DM1-usGNPs, reducing the IC_50_ value from 25.2 nM (95% CI: 21.5–29.7 nM) to 5 nM (95% CI: 4.2–5.9 nM) DM1 equivalent concentration, which is comparable to that of free DM1 in Hep3B cells and only moderately higher than the IC_50_ of free DM1 (1.4 nM; 95% CI: 1.2–1.6 nM) in U-87 MG cells. These data indicate that the presence of the cRGD peptide can be used to selectively target a DM1-loaded usGNP for cells expressing the αVβ3 integrin, providing a mechanism for the selective killing of cells and reducing off-target effects. 

## 4. Conclusions

Targeted delivery of cytotoxic drugs increases the therapeutic window by improving biodistribution, preventing off-target effects, or reducing drug degradation and elimination. Ultrasmall gold nanoparticles offer an improved surface-to-volume ratio that allows the immobilization of a larger number of different ligands at the same dose of gold, which presents an ideal platform for multiple functionalizations. Our strategy to link cRGD sidechains to the usGNP core via PEG(8)COOH lead to the successful binding of the functionalized usGNP to αVβ3 integrin both in vitro and in cell culture. This binding was selective and specifically proven by competition with free cRGD, functionalization with the non-interacting cRAD peptide, or using cell lines that do not express the αVβ3 integrin. The potential for multiple functionalization was shown by coupling fluorescent moieties to the usGNPs via the same PEG(8)COOH bridge, which permitted direct visualization of the usGNPs with confocal microscopy and monitoring their uptake and intracellular localization. Loading the maytansinoid drug DM1 onto the cRGD-usGNPs led to an improvement in selective toxicity on cells that express αVβ3 integrin and proved the potential of multi-functionalized usGNPs in improving the therapeutic window.

## Figures and Tables

**Figure 1 nanomaterials-12-04013-f001:**
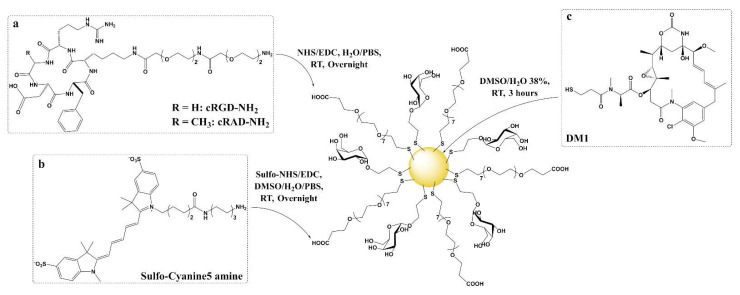
Four nanometers core usGNP platform (α-Galactose-C_2_)_145_(PEG(8)COOH)_145_@Au_2000_ composed of 2′-Thioethyl α-D-galactopyranoside and α-Thio-ω-(propionic acid) octa(ethylene glycol) synthesized with a modified Brust-Schiffrin method. (**a**) cRGD and cRAD amine derivatives and (**b**) Sulfo-Cyanine5 amine fluorophore moieties are bound through post-functionalization (amide coupling). (**c**) DM1 is linked by ligand exchange (direct anchoring to the core through a thiol group).

**Figure 2 nanomaterials-12-04013-f002:**
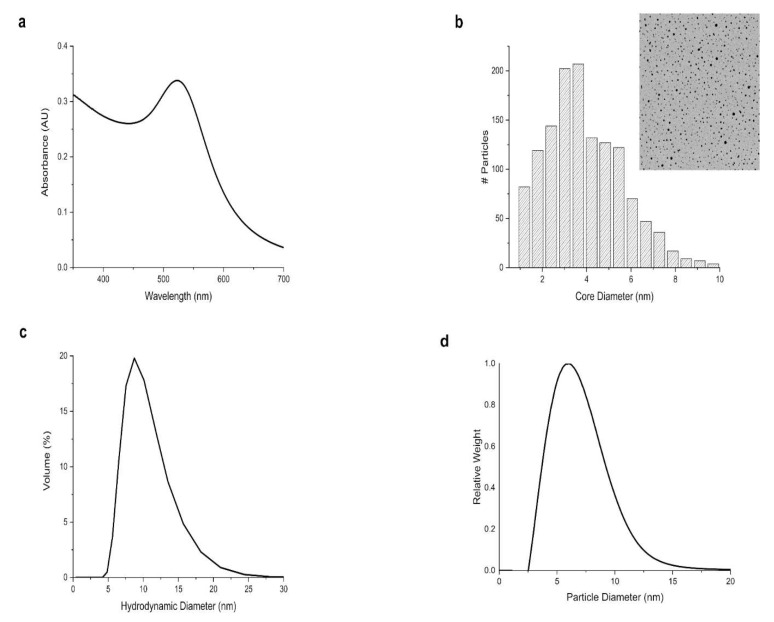
Characterization of the size and dispersity of usGNPs. (**a**) UV-Vis spectrum (λ 350–700 nm) shows a surface plasmon band with a local maximum at ~520 nm. (**b**) TEM picture and core size distribution indicate a mean core size of 4 nm. (**c**) DLS hydrodynamic size distribution by volume and (**d**) DCS diameter distribution confirm monodispersity and that functionalized particles remain in the ultrasmall range.

**Figure 3 nanomaterials-12-04013-f003:**
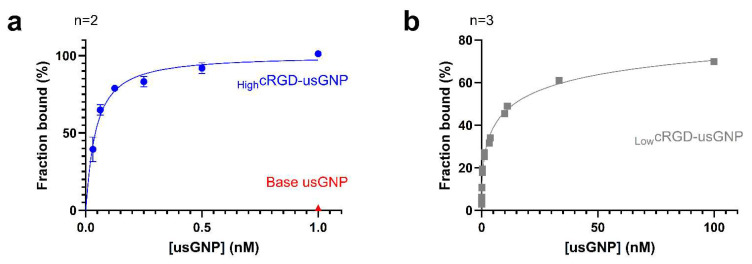
cRGD conjugated to usGNPs retain binding affinity to αVβ3 integrin. (**a**) Binding of _High_cRGD-usGNP to immobilized αVβ3 integrin. K_d_ = 29.2 ± 3.2 pM (*p* < 0.01; R^2^ = 0.991). No binding of the base usGNP was seen at the top concentration. (**b**) Binding of _Low_cRGD-usGNP to immobilized αVβ3 integrin. cRGD density significantly alters binding affinity. K_d_ = 3.2 ± 0.4 nM (*p* < 0.001; R^2^ = 0.995). Error bars represent standard deviation (SD) of n independent repeats (**a**,**b**). (**c**) Binding of cRGD-usGNPs is Ca^2+^-dependent: 0.25 nM _High_cRGD-usGNP was bound onto a biosensor with immobilized αVβ3 integrin in the presence of 50 µM Ca^2+^. Note the absence of binding and increased instability of the immobilized αVβ3 integrin dimer in the presence of 10 mM EDTA. (**d**) Binding onto the biosensor is cRGD-mediated: 0.25 nM cRGD-usGNP bound to immobilized αVβ3 integrin dissociates when 1 µM free cRGD is added as a competitor to the dissociation buffer. (**c**,**d**) Representative figures of n independent repeats.

**Figure 4 nanomaterials-12-04013-f004:**
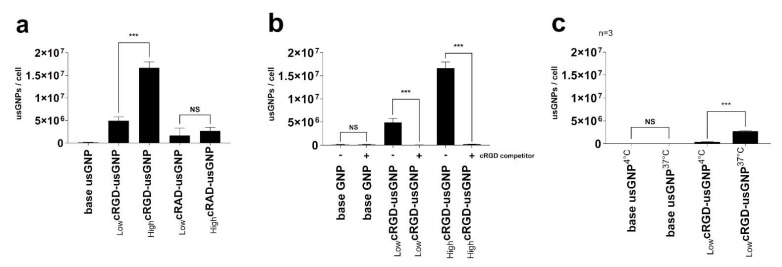
Uptake of usGNPs by U-87 MG cells is mediated via interaction with cRGD sidechains. Cells were incubated with the indicated usGNPs for 60 min then internalized gold was quantified with ICP-MS. (**a**) Uptake efficiency is dependent on the number of cRGD molecules loaded on the usGNPs. The cRAD low-binding peptide mediates only limited uptake by these cells. (**b**) Pre-treatment of cells with free cRGD peptides (+) interferes with the uptake of usGNPs, which confirms that uptake is αVβ3 integrin-mediated. (**c**) Uptake of usGNPs is an active process as incubating the cells with the cRGD-usGNPs at 4 °C results in the reduction of uptake. Error bars represent SD of n independent repeats. NS: *p* > 0.05; ***: *p* ≤ 0.001.

**Figure 5 nanomaterials-12-04013-f005:**
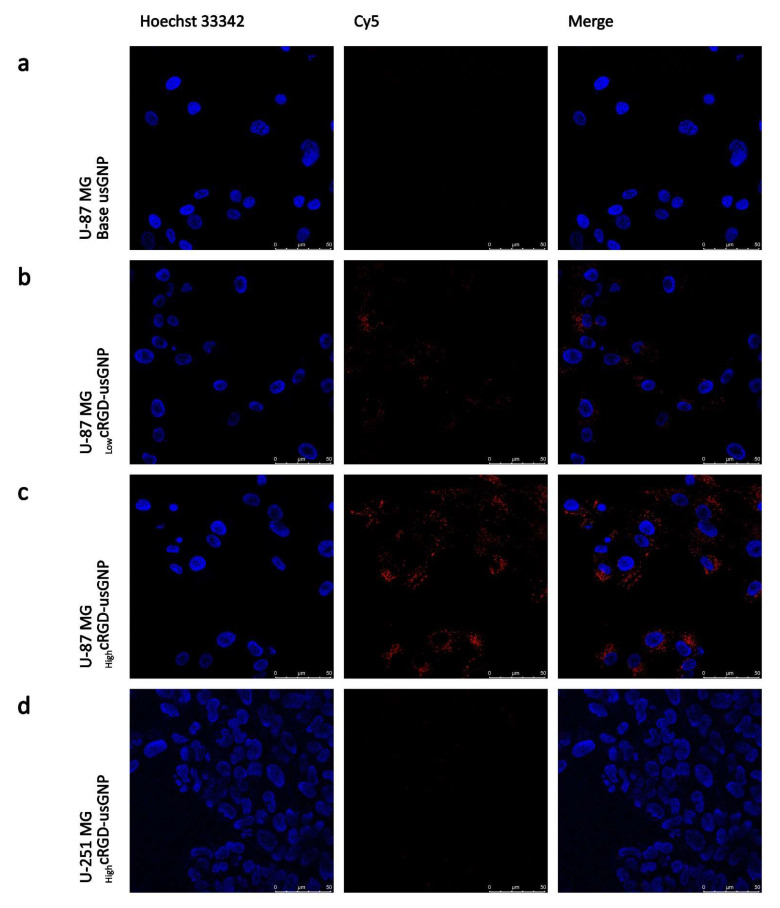
Uptake of usGNPs depends on αVβ3 integrin expression level. (**a**–**d**) U-87 MG and U-251 MG cells were incubated with 9 nM usGNPs with indicated cRGD densities for 2 h, then stained with 1 µg/mL Hoechst 33342 and imaged with live cell confocal microscopy. (**e**) Internalized usGNP content was quantified with ICP-AES following 2 h incubation. Representative figure from three (U87-MG) and one (U251-MG) independent repeats; error bars represent SD of five technical replicates.

**Figure 6 nanomaterials-12-04013-f006:**
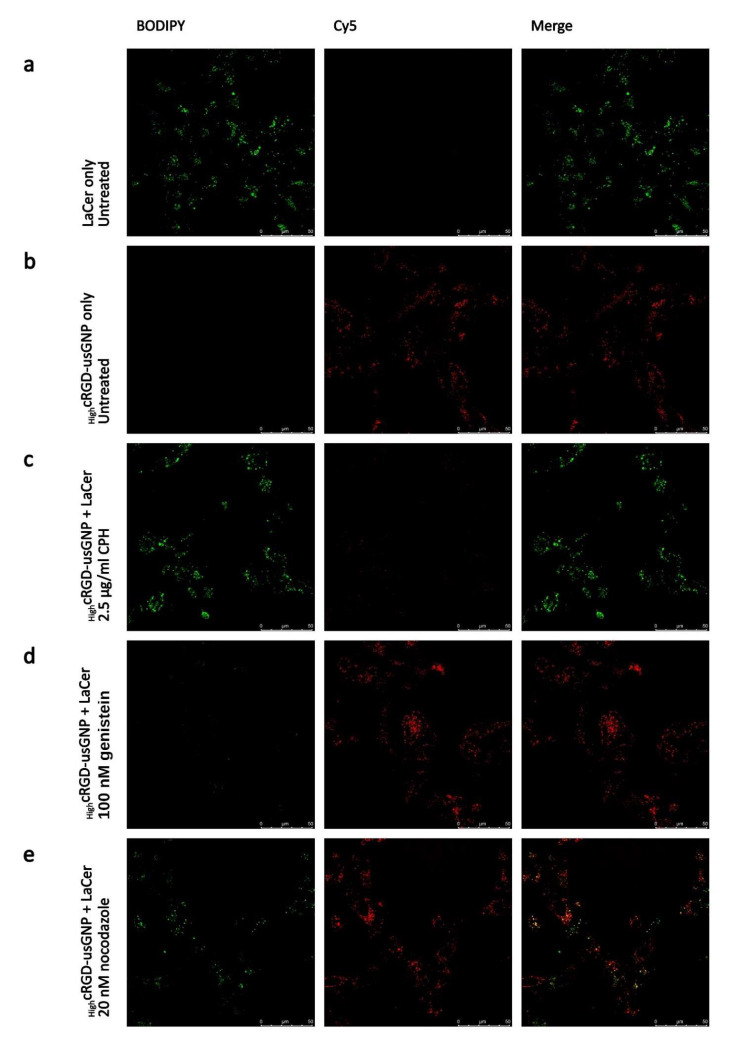
Inhibition of uptake of _High_cRGD-usGNPs and LaCer control in U-87 MG cells. (**a**–**e**) Cells were pre-treated with the indicated drugs for 30 min, then _High_cRGD-usGNPs at 9 nM final concentration and 1 μg/mL LaCer were added. Live cell images were acquired following a 2 h incubation, with filter settings optimized for BODIPY (LaCer, pseudo-colored green) and Cy5 (_High_cRGD-usGNPs, red).

**Figure 7 nanomaterials-12-04013-f007:**
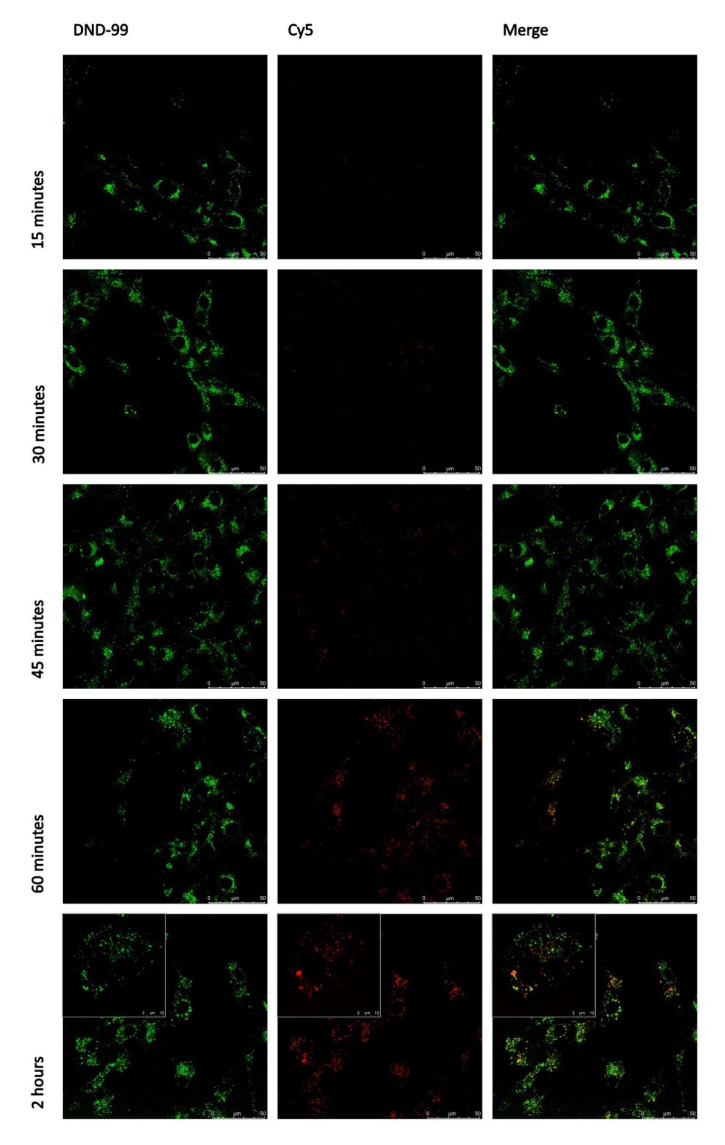
_High_cRGD-usGNPs accumulate in the lysosomes. U-87 MG cells were incubated with _High_cRGD-usGNPs at 9 nM final concentration for the indicated period. Fifteen minutes before acquisition 1:100 LysoTracker^®^ Red was added. Live cell images were taken with filter settings optimized for DND-99 (LysoTracker^®^ Red, pseudo-colored green) and Cy5 (_High_cRGD-usGNPs, red). The usGNPs appear co-localizing with the lysosomes after 30 min of incubation and reached a maximum after 2 h. Insets within the 2 h images show cells with a 3.5× zoom factor to better aid the comparison of co-localizing lysosomal and nanoparticle-derived signals.

**Figure 8 nanomaterials-12-04013-f008:**
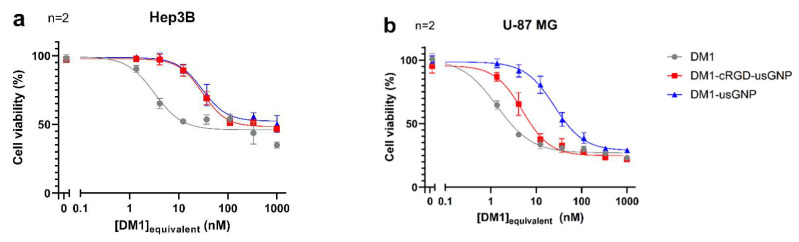
The presence of cRGD on usGNPs can selectively target αVβ3 integrin expressing cells with cytotoxic DM1 payload. αVβ3 integrin negative Hep3B (**a**) and positive U-87 MG (**b**) cells were treated with increasing concentrations of free DM1 (grey lines), DM1-functionalised base usGNP (blue) and DM1-_Low_cRGD-usGNPs (red). The presence of the cRGD ligand had no effect on the cytotoxicity of DM1 functionalized usGNPs in Hep3B cells. In U-87 MG cells, the cRGD sidechain increased cytotoxicity of DM1-_Low_cRGD-usGNPs nearly to the level of free DM1 confirming selective delivery of cytotoxic payload. Error bars represent SD of n independent experiments.

**Table 1 nanomaterials-12-04013-t001:** Chemical formula of the different usGNPs determined by ^1^H NMR and LC methods depicted using the (Ligands)_290_@Au_2000_ model [58], hydrodynamic diameter (volume distribution) as established by DLS and diameter by DCS (surface mode). N.D.: not determined.

usGNP	(Ligands)_290_@Au_2000_ Model	Hydrodynamic Diameter by DLS (mean; nm)	Diameter by DCS (mean; nm)
Base usGNP	(α-Galactose-C_2_)_145_(PEG(8)COOH)_145_@Au_2000_	9.1	9.7
_High_cRGD-usGNP	(α-Galactose-C_2_)_145_(PEG(8)COOH)_85_(cRGD)_60_@Au_2000_	11.4	10.1
_Low_cRGD-usGNP	(α-Galactose-C_2_)_145_(PEG(8)COOH)_125_(cRGD)_20_@Au_2000_	10.2	9.6
_High_cRAD-usGNP	(α-Galactose-C_2_)_145_(PEG(8)COOH)_85_(cRAD)_60_@Au_2000_	12.4	10.0
_Low_cRAD-usGNP	(α-Galactose-C_2_)_145_(PEG(8)COOH)_125_(cRAD)_20_@Au_2000_	11.5	9.9
DM1-cRGD-usGNP	(α-Galactose-C_2_)_135_(PEG(8)COOH)_115_(cRGD)_20_(DM1)_20_@Au_2000_	N.D.	N.D.
DM1-usGNP	(α-Galactose-C_2_)_135_(PEG(8)COOH)_135_(DM1)_20_@Au_2000_	N.D.	N.D.

## Data Availability

The data presented in this study are available on request from the corresponding author.

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
