# Peer review of "Targeting Ultrasmall Gold Nanoparticles with cRGD Peptide Increases the Uptake and Efficacy of Cytotoxic Payload"

_nanomaterials, 2022, doi:10.3390/nano12224013_

Round 1
Reviewer 1 Report
The article titled "Targeting ultrasmall gold nanoparticles to increase uptake and efficacy of cytotoxic payloads" presents a study investigating the increased cellular uptake of αVβ3 integrin-expressing cells by cRGD peptide-functionalized usGNPs. This is a potentially interesting study. However, there are many major points needed to be addressed:
1,The title is not appropriate, the author just use the cRGD modified the usGNP. We can’t see it in the title.
2, In line 286, Seeding 104 cells/well for 3 days is wrong, too many cells to accurately measure cell viability.
3,The authors synthesized 7 NPs in Table 1, but only characterized gold nanoparticles, we don’t know the diameter of the cRGD-usGNP.
4,In figure3, why is the number of repetitions sometimes 2 and sometimes 3?
5,In figure3 and 4, why the author didn’t test the cRAD-usGNP? I think cRAD-usGNP is also negative control.
6,The author should add a scale bar to the stained pictures.
7,U251-MG and U87-MG are GBM cells, they have too many differences besides the αVβ3 expression. The authors can’t get the conclusion just compare two cell lines. I suggest the authors KO the αVβ3 in U87 or overexpression the αVβ3 in U251.
8,I suggest the authors overexpression the αV and β3 at the same time, if β3 OE could induce the 293 cell apoptosis, it can be overexpressed in U251.
9,In figure7, the authors should enlarge the cells to better see the localization of NPs and lysosomes.
10, The IC50 of DM1-loaded usGNPs is higher than that of DM1, so it cannot be concluded that cRGD functionalization of DM1-loaded usGNPs improves cytotoxicity, the authors should load another drug. Or to test cytotoxicity in vivo, maybe NPs can enhance cytotoxicity.
11, Why the authors using the Hep3B cells to test the DM1-loaded usGNPs function?
Author Response
Please, see the attachment.

Reviewer 2 Report
In this manuscript, the authors developed a cRGD functionalized usGNPs that demonstrated increased cellular uptake by αVβ3 integrin expressing cells, are internalized via clathrin-dependent endocytosis, accumulate in the lysosomes, and when loaded with mertansine lead to increased cytotoxicity. In my opinion, this manuscript can be considered for publication in Nanomaterials after major revisions.
1. In the introduction section, the authors mentioned that the cyclic-peptide cRGD has previously been used to redirect drugs with five references. However, the paper (Protein Eng. Des. Sel. 2004, 17, 433-441) was published late, and I recommend to replace it to more recent references, such as Adv. Sci. 2018, 5, 1800581.
2. The dispersibility of nanoparticles is great important for their applications in biomedicine. Please provide the dynamic light scattering analysis of nanoparticles in different physiological solution at least for 7 days.
3. Please provide evidences that the drug DM1 was successfully loaded onto the ultra-small gold nanoparticles.
4. Can you try in vivo therapy? For evaluating the therapeutic efficiency, the in vivo combined therapy based on DM1 loaded ultra-small gold nanoparticles should be studied.
Round 2
Reviewer 1 Report
Accept in present form
Reviewer 2 Report
The manuscript is interesting and it can be accepted for publication in Nanomaterials.